# Meta-learning with differentiable closed-form solvers

**Luca Bertinetto**
FiveAI & University of Oxford
luca@robots.ox.ac.uk

**João Henriques**
University of Oxford
joao@robots.ox.ac.uk

**Philip H.S. Torr**
FiveAI & University of Oxford
philip.torr@eng.ox.ac.uk

**Andrea Vedaldi**
University of Oxford
vedaldi@robots.ox.ac.uk

## Abstract

Adapting deep networks to new concepts from a few examples is challenging, due to the high computational requirements of standard fine-tuning procedures. Most work on few-shot learning has thus focused on simple learning techniques for adaptation, such as nearest neighbours or gradient descent. Nonetheless, the machine learning literature contains a wealth of methods that learn non-deep models very efficiently. In this paper, we propose to use these fast convergent methods as the main adaptation mechanism for few-shot learning. The main idea is to teach a deep network to use standard machine learning tools, such as ridge regression, as part of its own internal model, enabling it to quickly adapt to novel data. This requires back-propagating errors through the solver steps. While normally the cost of the matrix operations involved in such a process would be significant, by using the Woodbury identity we can make the small number of examples work to our advantage. We propose both closed-form and iterative solvers, based on ridge regression and logistic regression components. Our methods constitute a simple and novel approach to the problem of few-shot learning and achieve performance competitive with or superior to the state of the art on three benchmarks.

## 1 Introduction

Humans can efficiently perform *fast mapping* (Carey, 1978; Carey & Bartlett, 1978), i.e. learning a new concept after a single exposure. By contrast, supervised learning algorithms — and neural networks in particular — typically need to be trained using a vast amount of data in order to generalize well. This requirement is problematic, as the availability of large labelled datasets cannot always be taken for granted. Labels can be costly to acquire: in drug discovery, for instance, campaign budgets often limits researchers to only operate with a small amount of biological data that can be used to form predictions about properties and activities of compounds (Altae-Tran et al., 2017). In other circumstances, data itself can be scarce, as it can happen for example with the problem of classifying rare animal species, whose exemplars are not easy to observe. Such a scenario, in which just one or a handful of training examples is provided, is referred to as *one-shot* or *few-shot learning* (Miller et al., 2000; Fei-Fei et al., 2006; Lake et al., 2015; Hariharan & Girshick, 2017) and has recently seen a tremendous surge in interest within the machine learning community (e.g.Vinyals et al. (2016); Bertinetto et al. (2016); Ravi & Larochelle (2017); Finn et al. (2017)).

Currently, most methods tackling few-shot learning operate within the general paradigm of *meta-learning*, which allows one to develop algorithms in which the process of learning can improve with the number of training episodes (Thrun, 1998; Vilalta & Drissi, 2002). This can be achieved by distilling and transferring knowledge across episodes. In practice, for the problem of few-shot classification, meta-learning is often implemented using two "nested training loops". The *base learner* works at the level of individual *episodes*, which correspond to learning problems characterised by having only a small set of labelled training images available. The *meta learner*, by contrast, learns from a collection of such episodes, with the goal of improving the performance of the base learner across episodes.

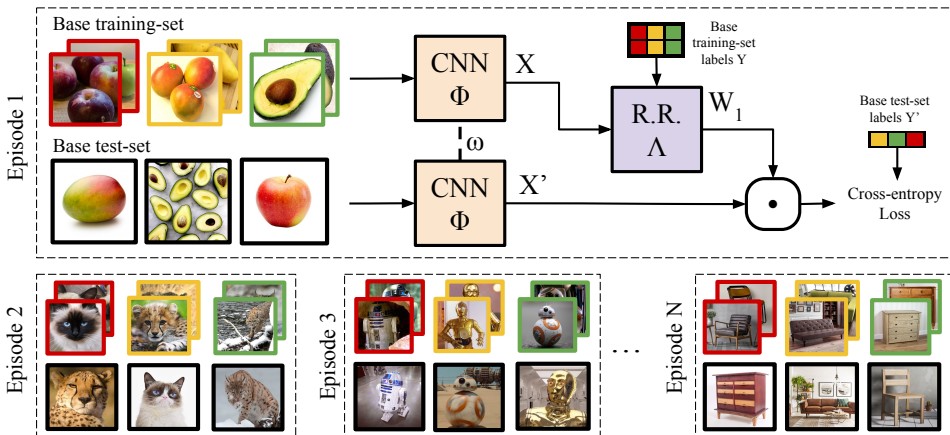

Figure 1: Diagram of the proposed method for one episode, of which several are seen during meta-training. The task is to learn *new* classes given just a few sample images per class. In this illustrative example, there are 3 classes and 2 samples per class, making each episode a 3-way, 2-shot classification problem. At the base learning level, learning is accomplished by a differentiable ridge regression layer (R.R.), which computes episode-specific weights (referred to as $w_{\mathcal{E}}$ in Section 3.1 and as $W$ in Section 3.2). At the meta-training level, by back-propagating errors through many of these small learning problems, we train a network whose weights are shared across episodes, together with the hyper-parameters of the R.R. layer. In this way, the R.R. base learner can improve its learning capabilities as the number of experienced episodes increases.

Clearly, in any meta-learning algorithm, it is of paramount importance to choose the base learner carefully. On one side of the spectrum, methods related to nearest-neighbours, such as learning similarity functions (Koch et al., 2015; Vinyals et al., 2016; Snell et al., 2017), are fast but rely solely on the quality of the similarity metric, with no additional data-dependent adaptation at test-time. On the other side of the spectrum, methods that optimize standard iterative learning algorithms, such as backpropagating through gradient descent (Finn et al., 2017; Nichol et al., 2018) or explicitly learning the learner's update rule (Hochreiter et al., 2001; Andrychowicz et al., 2016; Ravi & Larochelle, 2017), are slower but allow more adaptability to different problems/datasets.

In this paper, we take a different perspective. As base learners, we propose to adopt simple learning algorithms that admit a closed-form solution such as ridge regression. Crucially, the simplicity and differentiability of these solutions allow us to backpropagate through learning problems. Moreover, these algorithms are particularly suitable for use within a meta-learning framework for few-shot classification for two main reasons. First, their closed-form solution allows learning problems to be solved efficiently. Second, in a data regime characterized by few examples of high dimensionality, the Woodbury's identity (Petersen et al., 2008, Chapter 3.2) can be used to obtain a very significant gain in terms of computational speed.

We demonstrate the strength of our approach by performing extensive experiments on Omniglot (Lake et al., 2015), CIFAR-100 (Krizhevsky & Hinton, 2009) (adapted to the few-shot problem) and *mini*ImageNet (Vinyals et al., 2016). Our base learners are fast, simple to implement, and can achieve performance that is competitive with or superior to the state of the art in terms of accuracy.

## 2    RELATED WORK

The topic of meta-learning gained importance in the machine learning community several decades ago, with the first examples already appearing in the eighties and early nineties (Utgoff, 1986; Schmidhuber, 1987; Naik & Mammone, 1992; Bengio et al., 1992; Thrun & Pratt, 1998). Utgoff (1986) proposed a framework describing when and how it is useful to dynamically adjust the inductive bias of a learning algorithm, thus implicitly "changing the ordering" of the elements of its hypothesis space (Vilalta & Drissi, 2002). Later, Bengio et al. (1992) interpreted the update rule of a neural network's weights as a function that is learnable. Another seminal work is the one of Thrun (1996), which presents

the so-called *lifelong learning* scenario, where a learning algorithm gradually encounters an ordered sequence of learning problems. Throughout this course, the learner can benefit from re-using the knowledge accumulated during previous tasks. In later work, Thrun & Pratt (1998) stated that an algorithm is *learning to learn* if "[...] *its performance at each task improves with experience and with the number of tasks*". This characterisation has been inspired by Mitchell et al. (1997)'s definition of a learning algorithm as a computer program whose performance on a task improves with experience. Similarly, Vilalta & Drissi (2002) explained meta-learning as organised in two "nested learning levels". At the base level, an algorithm is confined within a limited hypothesis space while solving a single learning problem. Contrarily, the meta-level can "accrue knowledge" by spanning multiple problems, so that the hypothesis space at the base level can be adapted effectively.

Arguably, the simplest approach to meta-learning is to train a similarity function by exposing it to many matching problems (Bromley et al., 1993; Chopra et al., 2005; Koch et al., 2015). Despite its simplicity, this general strategy is particularly effective and it is at the core of several state-of-the-art few-shot classification algorithms (Vinyals et al., 2016; Snell et al., 2017; Sung et al., 2018). Interestingly, Garcia & Bruna (2018) interpret learning as information propagation from support (training) to query (test) images and propose a graph neural network that can generalize matching-based approaches. Since this line of work relies on learning a similarity metric, one distinctive characteristic is that parameter updates only occur within the long time horizon of the outer training loop. While this can clearly spare costly computations, it also prevents these methods from performing adaptation at test time. A possible way to overcome the lack of adaptability is to train a neural network capable of predicting (some of) its own parameters. This technique has been first introduced in Schmidhuber (1992; 1993) and recently revamped by Bertinetto et al. (2016) and Munkhdalai & Yu (2017). Rebuffi et al. (2017) showed that a similar approach can be used to adapt a neural network, on the fly, to entirely different visual domains.

Another popular approach to meta-learning is to interpret the gradient update of SGD as a parametric and learnable function rather than a fixed ad-hoc routine. Younger et al. (2001) and Hochreiter et al. (2001) observed that, because of the sequential nature of a learning algorithm, a recurrent neural network can be considered as a meta-learning system. They identify LSTMs as particularly apt for the task because of their ability to span long-term dependencies, which are essential in order to meta-learn. A modern take on this idea has been presented by Andrychowicz et al. (2016) and Ravi & Larochelle (2017), showing benefits on large-scale classification, style transfer and few-shot learning.

A recent and promising research direction is the one set by Maclaurin et al. (2015) and by the MAML algorithm (Finn et al., 2017; Finn & Levine, 2018). Instead of explicitly designing a meta-learner module for learning the update rule, they backpropagate through the very operation of gradient descent to optimize for the hyperparameters or the initial parameters of the learner. However, backpropagation through gradient descent steps is costly in terms of memory, and thus the total number of steps must be kept small.

To alleviate the drawback of catastrophic forgetting typical of deep neural networks (McCloskey & Cohen, 1989), several recent methods (Santoro et al., 2016; Kaiser et al., 2017; Munkhdalai & Yu, 2017; Sprechmann et al., 2018) make use of memory-augmented models, which can first retain and then access important and previously unseen information associated with newly encountered episodes. While such memory modules store and retrieve information in the long time range, approaches based on attention like the one of Vinyals et al. (2016) are useful to specify the most relevant pieces of knowledge within an episode. Mishra et al. (2018) complemented soft attention with temporal convolutions (Oord et al., 2016), thus allowing the attention mechanism to access information related to past episodes.

In this paper, we instead argue for simple, fast and differentiable base learners such as ridge regression. Compared to nearest-neighbour methods, they allow more flexibility because they produce a different set of parameters for different episodes ($W_i$ in Figure 1). Compared to methods that adapt SGD, they exhibit an inherently fast rate of convergence, particularly in cases where a closed form solution exists. A similar idea has been discussed by Bengio (2000), where the analytic formulations of zero-gradient solutions are used to obtain meta-gradients analytically and optimize hyper-parameters. More recently, Ionescu et al. (2015) and Valmadre et al. (2017) have derived backpropagation forms for the SVD and Correlation Filter, so that SGD can be applied, respectively, to a deep neural network that computes the solution to either an eigenvalue problem or a system of linear equations where the data matrix has a circulant structure.

## 3 METHOD

### 3.1 META-LEARNING

According to widely accepted definitions of learning (Mitchell, 1980) and meta-learning (Vilalta & Drissi, 2002; Vinyals et al., 2016), an algorithm is "learning to learn" if it can improve its learning skills with the number of experienced episodes (by progressively and dynamically modifying its inductive bias). There are two main components in a meta-learning algorithm: a base learner and a meta-learner (Vilalta & Drissi, 2002). The *base learner* works at the level of individual *episodes* (or *tasks*), which in the few-shot scenario correspond to learning problems characterised by having only a small set of labelled training images available. The *meta-learner* learns from several such episodes in sequence with the goal of improving the performance of the base learner across episodes.

In other words, the goal of meta-learning is to enable a base learning algorithm to adapt to new episodes efficiently by generalizing from a set of training episodes $\mathcal{E} \in \mathbb{E}$. $\mathcal{E}$ can be modelled as a probability distribution of example inputs $x \in \mathbb{R}^m$ and outputs $y \in \mathbb{R}^o$, such that we can write $(x, y) \sim \mathcal{E}$.

In the case of few-shot classification, the inputs are represented by few images belonging to different unseen classes, while the outputs are the (episode-specific) class labels. It is important not to confuse the small sets that are used in an episode $\mathcal{E}$ with the super-set $\mathbb{E}$ (such as Omniglot or *mini*ImageNet, Section 4.1) from which they are drawn.

Consider a generic feature extractor, such as commonly used pre-trained networks [1] $\phi(x) : \mathbb{R}^m \to \mathbb{R}^e$. Then, a much simpler episode-specific predictor $f(\phi(x); w_{\mathcal{E}}) : \mathbb{R}^e \times \mathbb{R}^p \to \mathbb{R}^o$ can be trained to map input embeddings to outputs. The predictor is parameterized by a set of parameters $w_{\mathcal{E}} \in \mathbb{R}^p$, which are specific to the episode $\mathcal{E}$.

To train and assess the predictor on one episode, we are given access to training samples $Z_{\mathcal{E}} = \{(x_i, y_i)\} \sim \mathcal{E}$ and test samples $Z'_{\mathcal{E}} = \{(x'_i, y'_i)\} \sim \mathcal{E}$, sampled independently from the distribution $\mathcal{E}$. We can then use a learning algorithm $\Lambda$ to obtain the parameters $w_{\mathcal{E}} = \Lambda(\phi(Z_{\mathcal{E}}))$, where $\phi(Z_{\mathcal{E}}) \triangleq \{(\phi(x_i), y_i)\}$. The expected quality of the trained predictor is then computed by a standard loss or error function $L : \mathbb{R}^o \times \mathbb{R}^o \to \mathbb{R}$, which is evaluated on the test samples $Z'_{\mathcal{E}}$:

$$q(\mathcal{E}) = \frac{1}{|Z'_{\mathcal{E}}|} \sum_{(x', y') \in Z'_{\mathcal{E}}} L\left(f\left(\phi\left(x'\right); w_{\mathcal{E}}\right), y'\right), \quad \text{with} \ w_{\mathcal{E}} = \Lambda(\phi(Z_{\mathcal{E}})). \quad (1)$$

Other than abstracting away the complexities of the learning algorithm as $\Lambda$, eq. (1) corresponds to the standard train-test protocol commonly employed in machine learning, here applied to a single episode $\mathcal{E}$. However, simply re-training a predictor for each episode ignores potentially useful knowledge that can be transferred between them. For this reason, we now take the step of parameterizing $\phi$ and $\Lambda$ with two sets of *meta-parameters*, respectively $\omega$ and $\rho$, which can aid the training procedure. In particular, $\omega$ affects the representation of the input of the base learner algorithm $\Lambda$, while $\rho$ corresponds to its hyper-parameters, which here can be learnt by the meta-learner loop instead of being manually set, as it usually happens in a standard training scenario. These meta-parameters will affect the generalization properties of the learned predictors. This motivates evaluating the result of training on a held-out test set $Z'_{\mathcal{E}}$ (eq. (1)). In order to learn $\omega$ and $\rho$, we minimize the expected loss on held-out test sets over all episodes $\mathcal{E} \in \mathbb{E}$:

$$\min_{\omega, \rho} \frac{1}{|\mathbb{E}| \cdot |Z'_{\mathcal{E}}|} \sum_{\mathcal{E} \in \mathbb{E}} \sum_{(x', y') \in Z'_{\mathcal{E}}} L\left(f\left(\phi\left(x'; \omega\right); w_{\mathcal{E}}\right), y'\right), \quad \text{with} \ w_{\mathcal{E}} = \Lambda(\phi(Z_{\mathcal{E}}; \omega); \rho). \quad (2)$$

Since eq. (2) consists of a composition of non-linear functions, we can leverage the same tools used successfully in deep learning, namely back-propagation and stochastic gradient descent (SGD), to optimize it. The main obstacle is to choose a learning algorithm $\Lambda$ that is amenable to optimization with such tools. This means that, in practice, $\Lambda$ must be quite simple.

**Examples of meta-learning algorithms.** Using eq. 2, it is possible to describe several of the meta-learning methods in the literature, which mostly differ for the choice of $\Lambda$. The feature extractor $\phi$ is typically a standard CNN, whose intermediate layers are trained jointly as $\omega$ (and thus are not

---

[1]Note that in practice we do not use pre-trained networks, but are able to train them from scratch.

episode-specific). The last layer represents the linear predictor $f$, with episode-specific parameters $w_{\mathcal{E}}$. In Siamese networks (Bromley et al., 1993; Chopra et al., 2005; Koch et al., 2015), $f$ is a nearest neighbour classifier, which becomes soft $k$-means in the semi-supervised setting proposed by Ren et al. (2018). Ravi & Larochelle (2017) and Andrychowicz et al. (2016) used an LSTM to implement $\Lambda$, while the Learnet (Bertinetto et al., 2016) uses a factorized CNN and MAML (Finn et al., 2017) implements it using SGD (and furthermore adapts all parameters of the CNN).

Instead, we use simple and fast-converging methods as base learner $\Lambda$, namely least-squares based solutions for ridge regression and logistic regression. In the outer loop, we allow SGD to learn both the parameters $\omega$ of the feature representation of $\Lambda$ and its hyper-parameters $\rho$.

## 3.2 Efficient ridge regression base learners

Similarly to the methods discussed in Section 3.1, over the course of a *single* episode we adapt a linear predictor $f$, which can be considered as the final layer of a CNN. The remaining layers $\phi$ are trained from scratch (within the outer loop of meta-learning) to generalize between episodes, but for the purposes of one episode they are considered fixed. In this section, we assume that the inputs were pre-processed by the CNN $\phi$, and that we are dealing only with the final linear predictor $f(\phi(x)) = \phi(x)W \in \mathbb{R}^o$, where the parameters $w_{\mathcal{E}}$ are reorganized into a matrix $W \in \mathbb{R}^{e \times o}$.

The motivation for our work is that, while not quite as simple as nearest neighbours, least-squares regressors admit closed-form solutions. Although simple least-squares is prone to overfitting, it is easy to augment it with $L^2$ regularization (controlled by a positive hyper-parameter $\lambda$), in what is known as ridge regression:

$$\Lambda(Z) = \underset{W}{\arg\min} \|XW - Y\|^2 + \lambda \|W\|^2 \tag{3}$$

$$= (X^T X + \lambda I)^{-1} X^T Y, \tag{4}$$

where $X \in \mathbb{R}^{n \times e}$ and $Y \in \mathbb{R}^{n \times o}$ contain the $n$ sample pairs of input embeddings and outputs from $Z$, stacked as rows.

Because ridge regression admits a closed form solution (eq. (4)), it is relatively easy to integrate into meta-learning (eq. (2)) using standard automatic differentiation packages. The only element that may have to be treated more carefully is the matrix inversion. When the matrix to invert is close to singular (which we do not expect when $\lambda > 0$), it is possible to achieve more numerically accurate results by replacing the matrix inverse and vector product with a linear system solver (Murphy, 2012, 7.5.2). In our experiments, the matrices were not close to singular and we did not find this necessary.

Another concern about eq. (4) is that the intermediate matrix $X^T X \in \mathbb{R}^{e \times e}$ grows *quadratically* with the embedding size $e$. Given the high dimensionality of features typically used in deep networks, the inversion could come at a very expensive cost. To alleviate this, we rely on the Woodbury formula (Petersen et al., 2008, Chapter 3.2), obtaining:

$$W = \Lambda(Z) = X^T (XX^T + \lambda I)^{-1} Y. \tag{5}$$

The main advantage of eq. (5) is that the intermediate matrix $XX^T \in \mathbb{R}^{n \times n}$ now grows quadratically with the number of samples in the episode, $n$. As we are interested in one or few-shot learning, this is typically very small. The overall cost of eq. (5) is only linear in the embedding size $e$.

Although this method was originally designed for regression, we found that it works well also in a (few-shot) classification scenario, where the target outputs are one-hot vectors representing classes. However, since eq. 4 does not directly produce classification labels, it is important to calibrate its output for the cross-entropy loss, which is used to evaluate the episode's test samples ($L$ in eq. 2). This can be done by simply adjusting our prediction $X'W$ with a scale and a bias $\alpha, \beta \in \mathbb{R}$:

$$\widehat{Y} = \alpha X'W + \beta. \tag{6}$$

Note that $\lambda$, $\alpha$ and $\beta$ are hyper-parameters of the base learner $\Lambda$ and can be learnt by the outer learning loop represented by the meta-learner, together with the CNN parameters $\omega$.

## 3.3 Iterative base learners and logistic regression

It is natural to ask whether other learning algorithms can be integrated as efficiently as ridge regression within our meta-learning framework. In general, a similar derivation is possible for iterative solvers,

as long as the operations are differentiable. For linear models with convex loss functions, a better choice than gradient descent is Newton's method, which uses curvature (second-order) information to reach the solution in very few steps. One learning objective of particular interest is logistic regression, which unlike ridge regression directly produces classification labels, and thus does not require the use of calibration before the (binary) cross-entropy loss.

When one applies Newton's method to logistic regression, the resulting algorithm takes a familiar form — it consists of a series of weighted least squares (or ridge regression) problems, giving it the name Iteratively Reweighted Least Squares (IRLS) (Murphy, 2012, Chapter 8.3.4). Given inputs $X \in \mathbb{R}^{n \times e}$ and binary outputs $y \in \{-1, 1\}^n$, the $i$-th iteration updates the parameters $w_i \in \mathbb{R}^e$ as:

$$w_i = \left(X^T \mathrm{diag}(s_i) X + \lambda I\right)^{-1} X^T \mathrm{diag}(s_i) z_i, \tag{7}$$

where $I$ is an identity matrix, $s_i = \mu_i(1 - \mu_i)$, $z_i = w_{i-1}^T X + (y - \mu_i)/s_i$, and $\mu_i = \sigma(w_{i-1}^T X)$ applies a sigmoid function $\sigma$ to the predictions using the previous parameters $w_{i-1}$.

Since eq. (7) takes a similar form to ridge regression, we can use it for meta-learning in the same way as in section 3.2, with the difference that a small number of steps (eq. (7)) must be performed in order to obtain the final parameters $w_{\mathcal{E}}$. Similarly, at each step $i$, we obtain a solution with a cost which is linear rather than quadratic in the embedding size by employing the Woodbury formula:

$$w_i = X^T \left(X X^T + \lambda \mathrm{diag}(s_i)^{-1}\right)^{-1} z_i,$$

where the inner inverse has negligible cost since it is a diagonal matrix. Note that a similar strategy could be followed for other learning algorithms based on IRLS, such as $L^1$ minimization and LASSO. We take logistic regression to be a sufficiently illustrative example, of particular interest for binary classification in one/few-shot learning, leaving the exploration of other variants for future work.

## 3.4 TRAINING POLICY

Figure 1 illustrates our overall framework. Like most meta-learning techniques, we organize our training procedure into *episodes*, each of which corresponds to a few-shot classification problem. In standard classification, training requires sampling from a distribution of images and labels. Instead, in our case we sample from a distribution of episodes, each containing its own training set and test set, with just a few samples per image. Each episode also contains two sets of labels: $Y$ and $Y'$. The former is used to train the base learner, while the latter to compute the error of the just-trained base learner, enabling back-propagation in order to learn $\omega$, $\lambda$, $\alpha$ and $\beta$.

In our implementation, one episode corresponds to a mini-batch of size $S = N(K + Q)$, where $N$ is the number of different classes ("ways"), $K$ the number of samples per classes ("shots") and $Q$ the number of query (or test) images per class.

## 4 EXPERIMENTS

In this section, we provide practical details for the two novel methods introduced in Section 3.2 and 3.3, which we dub R2-D2 (*Ridge Regression Differentiable Discriminator*) and LR-D2 (*Logistic Regression Differentiable Discriminator*). We analyze their performance against the recent literature on multi-class and binary classification problems using three few-shot learning benchmarks: Omniglot (Lake et al., 2015), *mini*ImageNet (Vinyals et al., 2016) and CIFAR-FS, which we introduce in this paper. The code for both our methods and the splits of CIFAR-FS are available at http://www.robots.ox.ac.uk/~luca/r2d2.html.

### 4.1 FEW-SHOT LEARNING BENCHMARKS

Let $I_\star$ and $C_\star$ be respectively the set of images and the set of classes belonging to a certain data split $\star$. In standard classification datasets, $I_{\mathrm{train}} \cap I_{\mathrm{test}} = \varnothing$ and $C_{\mathrm{train}} = C_{\mathrm{test}}$. Instead, the few-shot setup requires both $I_{\mathrm{meta\text{-}train}} \cap I_{\mathrm{meta\text{-}test}} = \varnothing$ and $C_{\mathrm{meta\text{-}train}} \cap C_{\mathrm{meta\text{-}test}} = \varnothing$, while within an episode we have $C_{\mathrm{task\text{-}train}} = C_{\mathrm{task\text{-}test}}$.

**Omniglot** (Lake et al., 2015) is a dataset of handwritten characters that has been referred to as the "MNIST transpose" for its high number of classes and small number of instances per class. It contains

20 examples of 1623 characters, grouped in 50 different alphabets. In order to be able to compare against the state of the art, we adopt the same setup and data split used in Vinyals et al. (2016). Hence, we resize images to $28\times28$ and we augment the dataset using four rotated versions of the each instance ($0°, 90°, 180°, 270°$). Including rotations, we use 4800 classes for meta-training and meta-validation and 1692 for meta-testing.

***mini*ImageNet** (Vinyals et al., 2016) aims at representing a challenging dataset without demanding considerable computational resources. It is randomly sampled from ImageNet (Russakovsky et al., 2015) and it is constituted by a total of 60,000 images from 100 different classes, each with 600 instances. All images are RGB and have been downsampled to $84\times84$. As all recent work, we adopt the same splits of Ravi & Larochelle (2017), who employ 64 classes for meta-training, 16 for meta-validation and 20 for meta-testing.

CIFAR-FS. On the one hand, despite being lightweight, Omniglot is becoming too simple for modern few-shot learning methods, especially with the splits of Vinyals et al. (2016). On the other, *mini*ImageNet is more challenging, but it might still require a model to train for several hours before convergence. Thus, we propose CIFAR-FS (CIFAR100 few-shots), which is randomly sampled from CIFAR-100 (Krizhevsky & Hinton, 2009) by using the same criteria with which *mini*ImageNet has been generated. We observed that the average inter-class similarity is sufficiently high to represent a challenge for the current state of the art. Moreover, the limited original resolution of $32\times32$ makes the task harder and at the same time allows fast prototyping.

## 4.2 Experimental results

In order to produce the features $X$ for the base learners (eq. 4 and 7), as many recent methods we use a shallow network of four convolutional "blocks", each consisting of the following sequence: a $3\times3$ convolution (padding=1, stride=1), batch-normalization, $2\times2$ max-pooling, and a leaky-ReLU with a factor of 0.1. Max pooling's stride is 2 for the first three layers and 1 for the last one. The four convolutional layers have $[96, 192, 384, 512]$ filters. Dropout is applied to the last two blocks for the experiments on *mini*ImageNet and CIFAR-FS, respectively with probabilities 0.1 and 0.4. We do not use any fully connected layer. Instead, we flatten and concatenate the output of the third *and* fourth convolutional blocks and feed it to the base learner. Doing so, we obtain high-dimensional features of size 3584, 72576 and 8064 for Omniglot, *mini*ImageNet and CIFAR-FS respectively. It is important to mention that the use of the Woodbury formula (section 3.2) allows us to make use of high-dimensional features without incurring burdensome computations. In fact, in few-shot problems the data matrix $X$ is particularly "large and short". As an example, with a 5-way/1-shot problem from *mini*ImageNet we have $X \in \mathbb{R}^{5\times72576}$. Applying the Woodbury identity, we obtain significant gains in computation, as in eq. 5 we invert a matrix that is only $5\times5$ instead of $72576\times72576$.

As Snell et al. (2017), we observe that using a higher number of classes during training is important. Hence, despite the few-shot problem at test time being 5 or 20-way, in our multi-class classification experiments we train using 60 classes for Omniglot, 16 for *mini*ImageNet and 20 for CIFAR-FS. Moreover, in order not to train a different model for every single configuration (two for *mini*ImageNet and CIFAR-FS, four for Omniglot), similarly to (Mishra et al., 2018) and differently from previous work, we train our models with a random number of shots, which does not deteriorate the performance and allow us to simply train one model per dataset. We then choose $Q$ (the size of the query or test set) accordingly, so that the batch size $S$ remains constant throughout the episodes. We set $S$ to 600 for Omniglot and 240 for both *mini*ImageNet and CIFAR-FS.

At the meta-learning level, we train our methods with Adam (Kingma & Ba, 2015) with an initial learning rate of 0.005, dampened by 0.5 every 2,000 episodes. Training is stopped when the error on the meta-validation set does not decrease meaningfully for 20,000 episodes.

As for the base learner, we let SGD learn the parameters $\omega$ of the CNN, as well as the regularization factor $\lambda$ and the scale $\alpha$ and bias $\beta$ of the calibration layer of R2-D2 (end of Section 3.2). In practice, we observed that it is important to use SGD to adapt $\alpha$ and $\beta$, while it is indifferent whether $\lambda$ is learnt or not. A more detailed analysis can be found in Appendix C.

**Multi-class classification.** Tables 1 and 2 show the performance of our closed-form base learner R2-D2 against the current state of the art for shallow architectures of four convolutional layers. Values represent average classification accuracies obtained by sampling 10,000 episodes from the

Table 1: Few-shot multi-class classification accuracies on *mini*ImageNet and CIFAR-FS.

| Method | *mini*ImageNet, 5-way | | CIFAR-FS, 5-way | |
|---|---|---|---|---|
| | 1-shot | 5-shot | 1-shot | 5-shot |
| MATCHING NET (Vinyals et al., 2016) | 44.2% | 57% | — | — |
| MAML (Finn et al., 2017) | 48.7±1.8% | 63.1±0.9% | 58.9±1.9% | 71.5±1.0% |
| MAML ∗ | 40.9±1.5% | 58.9±0.9% | 53.8±1.8% | 67.6±1.0% |
| META-LSTM (Ravi & Larochelle, 2017) | 43.4±0.8% | 60.6±0.7% | — | — |
| PROTO NET (Snell et al., 2017) | 47.4±0.6% | 65.4±0.5% | 55.5±0.7% | 72.0±0.6% |
| PROTO NET ∗ | 42.9±0.6% | 65.9±0.6% | 57.9±0.8% | 76.7±0.6% |
| RELATION NET (Sung et al., 2018) | 50.4±0.8% | 65.3±0.7% | 55.0±1.0% | 69.3±0.8% |
| SNAIL (with *ResNet*) (Mishra et al., 2018) | 55.7±*1.0%* | 68.9±*0.9%* | — | — |
| SNAIL (with 32C) (Mishra et al., 2018) | 45.1% | 55.2% | — | — |
| GNN (Garcia & Bruna, 2018) | 50.3% | 66.4% | 61.9% | 75.3% |
| GNN∗ | 50.3% | **68.2**% | 56.0% | 72.5% |
| OURS/R2-D2 (with 64C) | 49.5±0.2% | 65.4±0.2% | **62.3**±0.2% | **77.4**±0.2% |
| OURS/R2-D2 | **51.8**±0.2% | **68.4**±0.2% | **65.4**±0.2% | **79.4**±0.2% |
| OURS/LR-D2 (1 iter.) | 51.0±0.2% | 65.6±0.2% | **64.5**±0.2% | 75.8±0.2% |
| OURS/LR-D2 (5 iter.) | **51.9**±0.2% | **68.7**±0.2% | **65.3**±0.2% | 78.3±0.2% |

meta test-set and are presented with 95% confidence intervals. For each column, the best performance is in bold. If more than one value is outlined, it means their intervals overlap. For prototypical networks, we report the results reproduced by the code provided by the authors. For our comparison, we report the results of methods which train their models from scratch for few-shot classification, omitting very recent work of Qiao et al. (2018) and Gidaris & Komodakis (2018), which instead make use of pre-trained embeddings.

In terms of feature embeddings, Vinyals et al. (2016); Finn et al. (2017); Snell et al. (2017); Ravi & Larochelle (2017) use 64 filters per layer (which become 32 for *mini*ImageNet in (Ravi & Larochelle, 2017; Finn et al., 2017) to limit overfitting). On top of this, Sung et al. (2018) also uses a *relation module* of two convolutional and two fully connected layers. GNN (Garcia & Bruna, 2018) employs an embedding with $[64, 96, 128, 256]$ filters, a fully connected layer and a graph neural network (with its own extra parameters). In order to ensure a fair comparison, we increased the capacity of the architectures of three representative methods (MAML, prototypical networks and GNN) to match ours. The results of these experiments are reported with a ∗ on Table 1. We make use of dropout on the last two layers for all the experiments on baselines with ∗, as we verified it is helpful to reduce overfitting. Moreover, we report results for experiments on our R2-D2 in which we use a 64 channels embedding.

Despite its simplicity, our proposed method achieves an average accuracy that, on *mini*ImageNet and CIFAR-FS, is superior to the state of the art with shallow architectures. For example, on the four problems of Table 1, R2-D2 improves on average of a relative $4.3\%$ w.r.t. GNN (the second best method). R2-D2 shows competitive results also on Omniglot (Table 2), achieving among the best performance for all problems. Furthermore, when we use the "lighter" embedding, we can still observe a performance which is in line with the state of the art. Interestingly, increasing the capacity of the other methods it is not particularly helpful. It is beneficial only for GNN on *mini*ImageNet and prototypical networks on CIFAR-FS, while being detrimental in all the other cases.

Our R2-D2 is also competitive against SNAIL, which uses a much deeper architecture (a ResNet with a total of 14 convolutional layers). Despite being outperformed for the 1-shot case, we can match its results on the 5-shot one. Moreover, it is paramount for SNAIL to make use of such deep embedding, as its performance drops significantly with a shallow one.

**LR-D2 performance on multi-class classification.** In order to be able to compare our binary classifier LR-D2 with the state-of-the-art in few-shot $N$-class classification, it is possible to jointly consider $N$ binary classifiers, each of which discriminates between a specific class and all the remaining ones (Bishop, 2006, Chapter 4.1). In our framework, this can be easily implemented by concatenating together the outputs of N instances of LR-D2, resulting in a single multi-class prediction.

Table 2: Few-shot multi-class classification accuracies on Omniglot.

| Method | Omniglot, 5-way | | Omniglot, 20-way | |
|---|---|---|---|---|
| | 1-shot | 5-shot | 1-shot | 5-shot |
| SIAMESE NET (Koch et al., 2015) | 96.7% | 98.4% | 88% | 96.5% |
| MATCHING NET (Vinyals et al., 2016) | 98.1% | 98.9% | 93.8% | 98.5% |
| MAML (Finn et al., 2017) | **98.7**±0.4% | **99.9**±0.1% | 95.8±0.3% | 98.9±0.2% |
| PROTO NET (Snell et al., 2017) | 98.5±0.2% | 99.5±0.1% | 95.3±0.2% | 98.7±0.1% |
| SNAIL (Mishra et al., 2018) | **99.07**±0.16% | **99.77**±0.09% | **97.64**±0.30% | **99.36**±0.18% |
| GNN (Garcia & Bruna, 2018) | **99.2%** | **99.7%** | **97.4%** | 99.0% |
| OURS/R2-D2 (with 64C) | 98.55±0.05% | 99.66±0.02% | 94.70±0.05% | 98.91±0.02% |
| OURS/R2-D2 | **98.91**±0.05% | **99.74**±0.02% | 96.24±0.05% | **99.20**±0.02% |

Table 3: Few-shot binary classification accuracies on *mini*ImageNet and CIFAR-FS.

| Method | *mini*ImageNet, 2-way | | CIFAR-FS, 2-way | |
|---|---|---|---|---|
| | 1-shot | 5-shot | 1-shot | 5-shot |
| MAML (Finn et al., 2017) | 74.9±3.0% | 84.4±1.2% | 82.8±2.7% | 88.3±1.1% |
| PROTO NETS (Snell et al., 2017) | 71.7±1.0% | 84.8±0.7% | 76.4±0.9% | 88.5±0.6% |
| RELATION NET (Sung et al., 2018) | 76.2±1.2% | **86.8**±1.0% | 75.0±1.5% | 86.7±0.9% |
| GNN (Garcia & Bruna, 2018) | **78.4%** | **87.1%** | 79.3% | 89.1% |
| OURS/R2-D2 | 77.4±0.3% | **86.8**±0.2% | **84.1**±0.3% | **91.7**±0.2% |
| OURS/LR-D2 (10 iter.) | **78.1**±0.3% | 86.5±0.2% | **84.7**±0.3% | **91.5**±0.2% |

We use the same setup and hyper-parameters of R2-D2 (Section 4), except for the number of classes/ways used at training, which we limit to 10. Interestingly, with five IRLS iterations the accuracy of the 1-vs-all variant of LR-D2 is similar to the one of R2-D2 (Table 1): 51.9% and 68.7% for *mini*ImageNet (1-shot and 5-shot); 65.3% and 78.3% for CIFAR-FS. With a single iteration, performance is still very competitive: 51.0% and 65.6% for *mini*ImageNet; 64.5% and 75.8% for CIFAR-FS. However, the requirement of solving $N$ binary problems per iteration makes it much less efficient than R2-D2, as evident in Table 4.

**Binary classification.** Finally, in Table 3 we report the performance of both our ridge regression and logistic regression base learners, together with four representative methods. Since LR-D2 is limited to operate in a binary classification setup, we run our R2-D2 and prototypical network without oversampling the number of ways. For both methods and prototypical networks, we report the performance obtained annealing the learning rate by a factor of 0.99, which works better than the schedule used for multi-class classification. Moreover, motivated by the small size of the mini-batches, we replace Batch Normalization with Group Normalization (Wu & He, 2018). For this table, we use the default setup found in the code of MAML, which uses 5 SGD iterations during training and 10 during testing. Table 3 confirms the validity of both our approaches on the binary classification problem.

Although different in nature, both MAML and our LR-D2 make use of iterative base learners: the former is based on SGD, while the latter on Newton's method (under the form of Iteratively Reweighted Least Squares). The use of second-order optimization might suggest that LR-D2 is characterized by computationally demanding steps. However, we can apply the Woodbury identity at every iteration and obtain a significant speedup. In Figure 2 we compare the performance of LR-D2 vs the one of MAML for a different number of steps of the base learner (kept constant between training and testing). LR-D2 is superior to MAML, especially for a higher number of steps.

**Efficiency.** In Table 4 we compare the amount of time required by two representative methods and ours to solve 10,000 episodes (each with 10 images) on a single NVIDIA GTX 1080 GPU. We use *mini*ImageNet (5-way, 1-shot) and adopt, for the lower part of the table, a lightweight embedding network of 4 layers and 32 channels per layer. For reference, in the upper part of the table we also report the timings for R2-D2 with $[64, 64, 64, 64]$ and $[96, 192, 384, 512]$ embeddings.

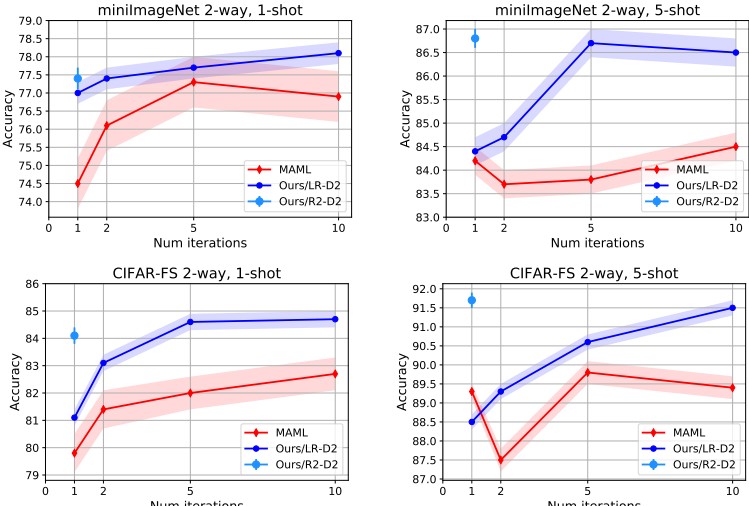

Figure 2: Binary classification accuracy on two datasets and two setups at different number of steps of the base learner for MAML, R2-D2 and LR-D2. Shaded areas represent 95% confidence intervals.

Interestingly, we can observe how R2-D2 allows us to achieve an efficiency that is comparable to the one of prototypical networks and significantly higher than MAML. Notably, unlike prototypical networks, our methods do allow per-episode adaptation through the weights $W$ of the solver.

Table 4: Time required to solve 10,000 *mini*ImageNet episodes of 10 samples each.

|  | *mini*ImageNet, 5-way, 1-shot |
|---|---|
| **OURS/R2-D2** | 1 min 23 sec |
| **OURS/R2-D2** (with 64C) | 1 min 4 sec |
| **MAML** (Finn et al., 2017) (with 32C) | 6 min 35 sec |
| **OURS/LR-D2** (1-vs-all) (1 iter.) (with 32C) | 5 min 48 sec |
| **OURS/R2-D2** (with 32C) | 57 sec |
| **PROTO NETS** (Snell et al., 2017) (with 32C) | 24 sec |

## 5 CONCLUSIONS

With the aim of allowing efficient adaptation to unseen learning problems, in this paper we explored the feasibility of incorporating fast solvers with closed-form solutions as the base learning component of a meta-learning system. Importantly, the use of the Woodbury identity allows significant computational gains in a scenario presenting only a few samples with high dimensionality, like one-shot of few-shot learning. R2-D2, the differentiable ridge regression base learner we introduce, is almost as fast as prototypical networks and strikes a useful compromise between not performing adaptation for new episodes (like metric-learning-based approaches) and conducting a costly iterative approach (like MAML or LSTM-based meta-learners). In general, we showed that our base learners work remarkably well, with excellent results on few-shot learning benchmarks, generalizing to episodes with new classes that were not seen during training. We believe that our findings point in an exciting direction of more sophisticated yet efficient online adaptation methods, able to leverage the potential of prior knowledge distilled in an offline training phase. In future work, we would like to explore Newton's methods with more complicated second-order structure than ridge regression.

### ACKNOWLEDGMENTS

We would like to thank Jack Valmadre, Namhoon Lee and the anonymous reviewers for their insightful comments, which have been useful to improve the manuscript. This work was partially supported by the ERC grant 638009-IDIU.

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

## A    EXTENDED DISCUSSION

**Contributions within the few-shot learning paradigm.** In this work, we evaluated our proposed methods R2-D2 and LR-D2 in the few-shot learning scenario (Fei-Fei et al., 2006; Lake et al., 2015; Vinyals et al., 2016; Ravi & Larochelle, 2017; Hariharan & Girshick, 2017), which consists in learning how to discriminate between images given one or very few examples. For methods tackling this problem, it is common practice to organise the training procedure in two *nested loops*. The inner loop is used to solve the actual few-shot classification problem, while the outer loop serves as a guidance for the former by gradually modifying the inductive bias of the base learner (Vilalta & Drissi, 2002). Differently from standard classification benchmarks, the few-shot ones enforce that classes are disjoint between dataset splits.

In the literature (e.g. Vinyals et al. (2016)), the very small classification problems with unseen classes solved within the inner loop have often been referred to as *episodes* or *tasks*. Considering the general few-shot learning paradigm just described, methods in the recent literature mostly differ for the type of learner they use in the inner loop and the amount of per-episode adaptability they allow. For example, at the one end of the spectrum in terms of "amount of adaptability", we can find methods such as MAML Finn et al. (2017), which learns how to efficiently fine-tune the parameters of a neural-network with few iterations of SGD. On the other end, we have methods based on metric learning such as prototypical networks Snell et al. (2017) and relation network Sung et al. (2018), which are fast but do not perform adaptation. Note that the amount of adaptation to a new episode (i.e.a new classification problem with unseen classes) is not at all indicative of the performance in few-shot learning benchmarks. As a matter of fact, both Snell et al. (2017) and Sung et al. (2018) achieve higher accuracy than MAML. Nonetheless, adaptability is a desirable property, as it allows more design flexibility.

Within this landscape, our work proposes a novel technique (R2-D2) that does allow per-episode adaptation while at the same time being fast (Table 4) and achieving strong performance (Table 1). The key innovation is to use a simple (and differentiable) solver such as ridge regression within the inner loop, which requires *back-propagating through the solution of a learning problem*. Crucially, its closed-form solution and the use of the Woodbury identity (particularly advantageous in the low data regime) allow this non-trivial endeavour to be efficient. We further demonstrate that this strategy is not limited to the ridge regression case, but it can also be extended to other solvers (LR-D2) by dividing the problem into a short series of weighted least squares problems ((Murphy, 2012, Chapter 8.3.4)).

**Disambiguation from the multi-task learning paradigm.** Our work – and more generally the few-shot learning literature as a whole – is related to the multi-task learning paradigm (Caruana, 1998; Ruder, 2017). However, several crucial differences exist. In terms of setup, multi-task learning methods are trained to solve a fixed set of $T$ tasks (or domains). At test time, the same $T$ tasks or domains are encountered. For instance, the popular Office-Caltech (Gong et al., 2012) dataset is constructed by considering all the images from 10 classes present in 4 different datasets (the domains). For multi-task learning, the splits span the domains but contain all the 10 classes. Conversely, few-shot learning datasets have splits with disjoint sets of classes (i.e. each split's classes are not contained in other splits). Moreover, only a few examples (*shots*) can be used as training data within one episode, while in multi-task learning this limitation is not present. For this reason, meta-learning methods applied to few-shot learning (e.g.ours, (Vinyals et al., 2016; Finn et al., 2017; Ravi & Larochelle, 2017; Mishra et al., 2018)) crucially take into account adaptation *already during the training process* to mimic the test-time setting, de facto learning how to learn from limited data.

**The importance of considering adaptation during training.** Considering adaptation during training is also one of the main traits that differentiate our approach from basic transfer learning approaches in which a neural network is first pre-trained on one dataset/task and then adapted to a different dataset/task by simply adapting the final layer(s) (e.g. Yosinski et al. (2014); Chu et al. (2016)).

To better illustrate this point, we conducted a baseline experiment. First, we pre-trained for a standard classification problem the same 4-layers CNN architecture using the same training datasets. We simply added a final fully-connected layer (with 64 outputs, like the number of classes in the training splits) and used the cross-entropy loss. Then, we used the convolutional part of this trained network as a feature extractor and fed its activations to our ridge-regression layer to produce a per-episode set of weights $W$. On miniImagenet, the drop in performance w.r.t. our proposed R2-D2 is very

significant: $-13.8\%$ and $-11.6\%$ accuracy for the 1 and 5 shot problems respectively. The drop in performance is consistent on CIFAR, though a bit less drastic: $-11.5\%$ and $-5.9\%$.

These results empirically confirm that simply using basic transfer learning techniques with a shared feature representation and task-specific final layers is *not* a good strategy to obtain results competitive with the state-of-the-art in few-shot learning. Instead, it is necessary to enforce the generality of the underlying features during training explicitly, which we do by back-propagating through the adaptation procedure (the regressors R2-D2 and LR-D2).

## B    DIFFERENT GAUSSIAN PRIORS FOR REGULARIZATION

The regularization term can be seen as a prior gaussian distribution of the parameters in a Bayesian interpretation, or more simply Tikhonov regularization (Tarantola, 2005). In the most common case of $\lambda I$, it corresponds to an isotropic gaussian prior on the parameters.

In addition to the case in which $\lambda$ is a scalar, we also experiment with the variant $\mathrm{diag}(\boldsymbol{\lambda})$, corresponding to an axis-aligned gaussian prior with an independent variance for each parameter, which can potentially exploit the fact that the parameters have different scales. Replacing $\lambda I$ with $\mathrm{diag}(\boldsymbol{\lambda})$ in 4, the final expression for W after having applied the Woodbury identity becomes:

$$W = \Lambda(Z) = \mathrm{diag}(\boldsymbol{\lambda})^{-1} X^T (X \mathrm{diag}(\boldsymbol{\lambda})^{-1} X^T + I)^{-1} Y. \tag{8}$$

## C    BASE LEARNER HYPER-PARAMETERS

Figure 3 illustrates the effect of using SGD to learn, together with the parameters $\omega$ of the CNN, also the hyper-parameters ($\rho$ in eq. 2) of the base learner $\Lambda$. We find that it is very important to learn the scalar $\alpha$ (right plot of Figure 3) used to calibrate the output of R2-D2 in eq. 6, while it is indifferent whether or not to learn $\lambda$. Note that, by using SGD to update $\alpha$, it is possible (e.g.in the range $[10^{-3}, 10^0]$) to recover from poor initial values and suffer just a little performance loss w.r.t. the optimal value of $\alpha = 10$.

The left plot of Figure 3 also shows the performance of R2-D2 with the variant $\mathrm{diag}(\boldsymbol{\lambda})$ introduced in Appendix B. Unfortunately, despite this formulation allows us to make use of a more expressive prior, it does not improve the results compared to using a simple scalar $\lambda$. Moreover, performance abruptly deteriorate for $\lambda > 0.01$.

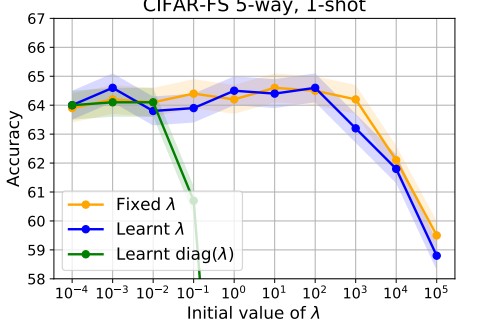 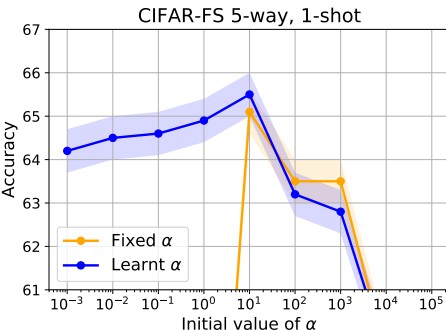

Figure 3:  Shaded areas represent 95% confidence intervals.

