# OpenReview forum: "Meta-learning with differentiable closed-form solvers"
_ICLR.cc/2019/Conference_

### Official Review · AnonReviewer2 · 2018-11-02
**A good idea that achieves good results**

**Rating:** 7
**Confidence:** 4

**Review:**

This paper proposes a meta-learning approach for the problem of few-shot classification. Their method, based on parametrizing the learner for each task by a closed-form solver, strikes an interesting compromise between not performing any adaptation for each new task (as is the case in pure metric learning methods) and performing an expensive iterative procedure, such as MAML or Meta-Learner LSTM where there is no guarantee that after taking the few steps prescribed by the respective algorithms the learner has converged. For this reason, I find that leveraging existing solvers that admit closed-form solutions is an attractive and natural choice.

Specifically, they propose ridge regression as their closed-form solver (R2-D2 variant). This is easily incorporated into the meta-learning loop with any hyperparameters of this solver being meta-learned, along with the embedding weights as is usually done. The use of the Woodbury equation allows to rewrite the closed form solution in a way that scales with the number of examples instead of the dimensionality of the features; therefore taking advantage of the fact that we are operating in a few-shot setting. While regression may seem to be a strange choice for eventually solving a classification task, it is used as far as I understand due to the availability of this widely-known closed-form solution. They treat the one-hot encoded labels of the support set as the regression targets, and additionally calibrate the output of the network (via a transformation by a scale and bias) in order to make it appropriate for classification. Based on the loss of ridge regression on the support set of a task, a parameter matrix is learned for that task that maps from the embedding dimensionality to the number of classes. This matrix can then be used directly to multiply the embedded (via the fixed for the purposes of the episode embedding function) query points, and for each query point, the entry with the maximum value in the corresponding row of the resulting matrix will constitute the predicted class label.

They also experimented with a logistic regression variant (LR-D2) that does not admit a closed-form solution but can be solved efficiently via Newton’s Method under the form of Iteratively Reweighted Least Squares. When using this variant they restrict to tackling the case of binary-classification.

A question that comes to mind about the LR-D2 variant: while I understand that a single logistic regression classifier is only capable of binary classification, there seems to be a straightforward extension to the case of multiple classes, where one classifier per class is learned, leading to a total of N one-vs-all classifiers (where N is the way of the episode). I’m curious how this would compare in terms of performance against the ridge regression variant which is naturally multi-class. This would allow to directly apply this variant in the common setting and would enable for example still oversampling classes at meta-training time as is done usually.

I would also be curious to see an ablation where for the LR-D2 variant SGD was used as the optimizer instead of Newton’s method. That variant may require more steps (similar to MAML), but I’m curious in practice how this performs.

A few other minor comments:
- In the related work section, the authors write: “On the other side of the spectrum, methods that optimize standard iterative learning algorithms, [...] are accurate but slow.” Note however that neither MAML nor MetaLearner LSTM have been showed to be as effective as Prototypical Networks for example. So I wouldn’t really present this as a trade-off between accuracy and speed.
- I find the term multinomial classification strange. Why not use multi-class classification?
- In page 8, there is a sentence that is not entirely grammatically correct: ‘Interestingly, increasing the capacity of the other method it is not particularly helpful’.

Overall, I think this is good work. The idea is natural and attractive. The writing is clear and comprehensive. I enjoyed how the explanation of meta learning and the usual episodic framework was presented. I found the related work section thorough and accurate too. The experiments are thorough as well, with appropriate ablations to account for different numbers of parameters used between different methods being compared. This approach is evidently effective for few-shot learning, as demonstrated on the common two benchmarks as well as on a newly-introduced variant of cifar that is tailored to few-shot classification. Notably, the ridge regression variant can reach results competitive with SNAIL that uses significantly more weights and is shown to suffer when its capacity is reduced. Interestingly, other models such as MAML actually suffer when given additional capacity, potentially due to overfitting.

---

> ### Author Response · Authors · 2018-11-07
> **Response to AR2**
>
> We thank the reviewer for the insightful comments and analysis.
>
> > “One-vs-all classifiers” for LR-D2
> This is a great suggestion, and we are not quite sure how we missed it. We will update the results for 5-way classification incorporating this method.
>
> > “ablation where for the LR-D2 variant SGD was used ... instead of Newton’s method”
> We previously did exactly this experiment, although for the R2-D2 (ridge regression) variant. We did not include it due to space constraints. It is equivalent to MAML, which also uses SGD, but adapting only the classification layer for new tasks (instead of adapting all parameters).
>
> We tested this variant on miniImageNet with 5 classes, with the lowest-capacity CNN (which is the most favorable model for MAML/SGD). It yields 45.4±1.6% accuracy for 1-shot classification and 61.7±1.0% for 5-shot classification. Comparing it to Table 1, there’s a drop in performance compared to our closed form solver (3.5% and 4.4% less accuracy, respectively), and also compared to the original MAML (3.3% and 1.4% respectively).
>
> Although we expect the conclusions for logistic regression (LR-D2) to be similar, we will extend the experiment to this case and report the results.
>
> > “Neither MAML nor MetaLearner LSTM have been showed to be as effective as Prototypical Networks for example”
> We agree, and will amend the text. Their interest may lie more in their technical novelty.
>
> > Suggestions on multinomial term and sentence grammar
> These do improve the readability of the text and will be corrected.

---

### Official Review · AnonReviewer1 · 2018-11-03
**Not clear what is novel here**

**Rating:** 2
**Confidence:** 5

**Review:**

Summary: The paper proposes an algorithm for meta-learning which amounts to fixing the features (ie all hidden layers of a deep NN), and treating each task  as having its own final layer which could be a ridge regression or a logistic regression. The paper also proposes to separate the data for each task into a training set used to optimize the last, task specific layer, and a validation set used to optimize all previous layers and hyper parameters.

Novelty: This reviewer is unsure what the paper claims as a novel contribution. In particular training multi-task neural nets with shared feature representation and task specific final layer is probably 20-30 years old by now and entirely common. It is also common freeze the feature representation learned from the first set of tasks, and to simply use it for new tasks by modifying the last (few) layer(s) which would according to this paper qualify as meta-learning since the new task can be learned with very few new examples.

---

> ### Author Response · Authors · 2018-11-13
> **This is a comment on a different technique than what we propose**
>
> We thank the reviewer for the comment.
> However, we believe that the low score originates from a misunderstanding of our proposal.
> Below, we try to bring some clarity by disambiguating between what the reviewer refers to and our method.
> If our interpretation of what the reviewer refers to as “entirely common” is incorrect, it would be great to be provided with at least one reference, so that we can continue the conversation on the same ground.
>
> > “novel contribution?” , “training multi-task neural nets with shared feature representation and task specific final layer is probably 20-30 years old by now and entirely common.”
> “It is also common freeze the feature representation learned from the first set of tasks, and to simply use it for new tasks by modifying the last layer”
>
> We understand that the reviewer is hinting at the common multi-task scenario with a shared network and task-specific layers (e.g. Caruana 1993). He/she also refers to basic transfer learning approaches in which a CNN is first pre-trained on one dataset/task and then adapted to a different dataset/task by simply adapting the final layer(s) (e.g. Yosinski et al. “How transferable are features in deep neural Networks?” - NIPS 2014; Chu et al. “Best Practices for Fine-tuning Visual Classifiers to New Domains” - ECCVw 2016).
>
> If so, then this is significantly different to our work, and in general to all of the previous literature on meta-learning applied to few-shot classification (e.g. Finn et al. 2017, Ravi & Larochelle 2017, Vinyals et al. 2016, etc).
> Notably, these methods and ours take into account adaptation *already during the training process*, which requires back-propagating errors through the very fine-tuning process.
>
> Within this setup, our main contribution is to propose an adaptation procedure based on closed-form regressors, which have the important characteristic of allowing different models for different episodes while still being fast because of 1) their convergence in one (R2-D2) or few (LR-D2) steps, 2) the use of the Woodbury identity, which is particularly convenient in the few-shot data regime, and 3) back-propagation through the closed-form regressor can be made efficient.
>
> To better illustrate our point, we conducted a baseline experiment.
> First, we pre-trained the same 4-layers CNN architecture, but for a standard classification problem, using the same training samples as our method. We simply added a final fully-connected layer (with 64 outputs, like the number of classes in the training splits) and used the cross-entropy loss.
> Then, we used the convolutional part of this trained network as a feature extractor and fed its activation to our ridge-regression layer to produce a per-episode set of weights.
> On miniImagenet, the drop in performance w.r.t. our proposed R2-D2 is very significant: 13.8% and 11.6% accuracy for the 1 and 5 shot problems respectively.
> Results are consistent on CIFAR, though less drastic: 11.5% and 5.9%.
>
> This confirms that simply using a “shared feature representation and task specific final layer” as commented by the reviewer is not what we are doing and it is not a good strategy to obtain results competitive with the state-of-the-art in few-shot classification.
> Instead, it is necessary to enforce the generality of the underlying features during training explicitly, which we do by back-propagating through the fine-tuning procedure (the closed-form regressors).
>
> We would like to conclude remarking that, probably, the source of confusion arises from the overlap that exists in general between the few-shot learning and the transfer/multi-task learning sub-communities.
> We realize that the two have developed fairly separately while trying to solve very related problems, and unfortunately the similarities/differences are not acknowledged enough in few-shot classification papers, including our own. We intend to alleviate this problem in our related work section, and invite the reviewer to suggest more relevant works from this area.

---

> > ### Comment · AnonReviewer1 · 2018-12-08
> > **I still feel the novelty is very small (I'm reviewer #1)**
> >
> > I disagree with Reviewer #2 and the authors about the novelty. The delta from just simple multi-task learning approach of eg Caruana 93 is extremely small -- the same algorithms are trivially extended to deal with meta-learning. The mere fact of using closed form ridge regression in this setting does not feel like sufficient contribution to warrant an ICLR paper to this reviewer. Merely because some other paper also had small novelty and got accepted in the past I can not see why this paper should also get accepted with minimal novel contributions.

---

> > > ### Author Response · Authors · 2018-12-09
> > > **3 papers with hundreds of citations (Finn et al., Ravi & Larochelle, Vinyals et al.) cannot be dismissed as “some other paper also had small novelty”**
> > >
> > > The reviewer has not refuted any of the points we made above. Namely:
> > >
> > > - That meta-learning approaches (like ours) back-propagate errors through the fine-tuning process, a major departure from standard multi-task/transfer learning.
> > > - That *not doing so* incurs a large performance penalty, as demonstrated by our experiments.
> > >
> > > We invite the reviewer to address these points, rather than just reiterate a subjective judgment over the value of meta-learning. While we respect this opinion, our paper cannot be rejected based solely on the reviewer’s opinion that meta-learning papers are not novel in general (compared to multi-task learning).

---

> > > > ### Public Comment · (anonymous) · 2018-12-13
> > > > **It is misleading to indicate that the paper is as novel as the 3 meta-learning papers**
> > > >
> > > > The 3 meta-learning papers developed new techniques and/or models for meta-learning (which have never been proposed or used in multi-task learning), while this paper applies existing multi-task learning technique in the meta-learning setting. The contributions in these two cases are very different. I think it is misleading to indicating that the contribution of this paper is as novel as the 3 meta-learning papers.
> > > >
> > > > It is fine to apply multi-task learning technique to the meta-learning problem. To some extent, meta-learning can be explained as a generalization of multi-task learning in the way that meta-learning applies to any set of tasks sampled from certain *task distribution*, while the set of tasks in multi-task learning are fixed. They both need knowledge transfer between different tasks. However, using a multi-task technique in meta-learning setting cannot be treated as a novel or original contribution.

---

> > > > > ### Author Response · Authors · 2018-12-13
> > > > > **We were responding to the claim that these papers had small novelty and thus our own as well, which we disagree with**
> > > > >
> > > > > 1) We wrote: ““[multi-task learning] is different to our work, and in general to all of the previous literature on meta-learning applied to few-shot classification (e.g. Finn et al. 2017, Ravi & Larochelle 2017, Vinyals et al. 2016, etc). Notably, these methods and ours take into account adaptation *already during the training process*, which requires back-propagating errors through the very fine-tuning process.””
> > > > >
> > > > > 2) R1 answered with: ““Merely because some other paper also had small novelty and got accepted in the past I can not see why this paper should also get accepted””
> > > > >
> > > > > 3) We then observed that *R1 did not refute any of our point of rebuttal* (long answer in this thread) and seems to be dismissive of the above papers, which are widely accepted by the community.
> > > > >
> > > > > > ““ However, using a multi-task technique in meta-learning setting cannot be treated as a novel or original contribution.””
> > > > > Again, it is not what we do - we amply addressed this point both on OpenReview (last two answers to the reviewer) and in the paper.
> > > > >
> > > > > We would like to repeat that if this were true, the baseline experiment we described (applying ridge regression in the manner that the reviewer refers to as standard) would not have been possible, since our method and the baseline would then be the same (which they are not -- both in methodology and results).

---

### Official Review · AnonReviewer3 · 2018-11-05
**results are not very promising**

**Rating:** 5
**Confidence:** 3

**Review:**

This paper proposes a new meta-learning method based on closed-form solutions for task specific classifiers such as ridge regression and logistic regression (iterative). The idea of the paper is quite interesting, comparing to the existing metric learning based methods and optimization based methods.

I have two concerns on this paper.
First, the motivation and the rationale of the proposed approach is not clear. In particular, why one can simply treat \hat{Y} as a scaled and shifted version of X’W?

Second, the empirical performance of the proposed approach is not very promising and it does not outperform the comparison methods, e.g., SNAIL.  It is not clear what is the advantage.

---

> ### Author Response · Authors · 2018-11-08
> **Our proposal demonstrates results competitive with SNAIL despite using a much simpler architecture (SNAIL uses ResNet, we just use 4 conv layers).**
>
> We thank the reviewer for the comments and questions.
>
> > “Why one can simply treat \hat{Y} as a scaled and shifted version of X’W?”
> In the case of logistic regression, the scaling and shifting is not needed, and we have \hat{Y}=X’W. This is because logistic regression is a classification algorithm, and directly outputs class scores. These scores are fed to the (cross-entropy) loss L.
>
> However, ridge regression is a regression algorithm, and its regression targets are one-hot encoded labels, which is only an approximation of the discrete problem (classification). This means that an extra calibration step is needed (eq. 6), to allow the network to tune the regressed outputs into classification scores for the cross-entropy loss L.
>
> > “The empirical performance of the proposed approach is not very promising and it does not outperform the comparison methods, e.g., SNAIL”
> Our method actually outperforms SNAIL on an apples-to-apples comparison, with the same number of layers. We would like to draw the reviewer’s attention to the last paragraph of the “Multi-class classification” subsection (page 8).
>
> The result mentioned by the reviewer uses a ResNet, while we use a 4-layer CNN to remain comparable to prior work. SNAIL with a 4-layer CNN ([11] Appendix B) performs much worse than our method (7.4% to 10.0% accuracy improvement).
>
> Even disregarding the great difference in architecture capacity, our proposal's performance coincides with SNAIL on miniImageNet 5way-5shot and it is comparable on 3 out of 4 Omniglot setups. We would have liked to establish a comparison also on CIFAR, but unfortunately the official code for SNAIL hasn’t been released.
>
> Borrowing the words of AnonReviewer2: “Notably, the ridge regression variant can reach results competitive with SNAIL that uses significantly more weights and is shown to suffer when its capacity is reduced.”
>
> We hope that this addresses the two concerns raised by the reviewer. We will be happy to answer any other question about the paper.

---

> > ### Public Comment · (anonymous) · 2018-12-06
> > **interesting idea but underwhelming results**
> >
> > Using closed-form base learner is an interesting idea. However, the results are underwhelming.
> >
> > As shown in https://github.com/gidariss/FewShotWithoutForgetting , Prototypical Networks can be quite powerful with some modifications. The modifications include:
> > 1. add data augmentation
> > 2. use SGD with momentum optimizer
> > 3. scale the output of the euclidean distance to a suitable range
> >
> > Using a 4-Conv backbone with 64 channels, Protypical Networks are able achieve remarkable results in MiniImagenet:  1-shot: 53.30% +/- 0.79 5-shot: 70.33% +/- 0.65
> >
> > Even without data augmentation, in my experiments, Protypical Networks can still get 5-shot accuracy around 68.8%.
> >
> > Considering this, the proposed method has not demonstrated superior empirical results than Protypical Network yet.

---

> > > ### Author Response · Authors · 2018-12-06
> > > **These improvements can be used in any few-shot learning methods. We outperform prototypical networks in an apples-to-apples comparison.**
> > >
> > > We thank the anonymous commenter for pointing out a GitHub repo with improvements. We note that neither data augmentation nor the optimizer schedule are mentioned at all in the associated published paper.
> > >
> > > Additionally, the mentioned improvements are not specific to prototypical networks (or to any method for that matter), and can also be applied to ours.  As such, we fail to see how this says anything about the merits of our proposal.
> > > In our experiments, we compare against prototypical networks using the same setup of the original paper (Adam optimizer, halving LR every 20 epochs; no data augmentation).
> > > In this fair comparison, we outperform it.
> > >
> > > We would gain no knowledge by showing that “proto-nets with data augmentation and optimizer improvements” (as suggested) beats “R2D2 with no data augmentation”, or that “MAML with a ResNet base” beats “R2D2 with 4 layers”. These are apples-to-oranges comparisons which make any scientific conclusion very hard to draw.
> > >
> > > Instead, a proper comparison is to take the innovation of each paper -- the prototype layer in proto-nets, and the ridge regression layer in R2D2 -- and compare them, with everything else fixed. This includes data augmentation, as well as network model and initialization.
> > >
> > > Carefully controlled comparisons are a core part of the scientific method, and ignoring them will lead to unsubstantiated conclusions.

---

> > > > ### Public Comment · (anonymous) · 2018-12-07
> > > > **what happens when the backbone network gets deeper**
> > > >
> > > > I understand your points. Overall, I like the idea of using closed-form base learner which also demonstrate good performance when the backbone network is shallow. However, as a practioner, I may not adopt the proposed method for now.
> > > >
> > > > In my opinion, meta-learning is about learning a data-driven inductive bias for few-shot learning. Closed-form regression itself introduces a strong inductive bias which is not learned. Therefore, it is interesting to investigate whether the inductive bias of closed-form regression is needed when the backbone network gets deeper.
> > > >
> > > > As shown in the Figure 3 of https://openreview.net/pdf?id=HkxLXnAcFQ , the performance gap between different meta-learning methods diminishes as the backbone gets deeper. One intersting point in the figure is that ProtoNet typically outperforms other methods when the network is deeper.

---

> > > > > ### Author Response · Authors · 2018-12-07
> > > > > **Thanks**
> > > > >
> > > > > Thank you for pointing us to this interesting paper! We agree that methods with limited inductive bias such as protonets are attractive, and there is indeed a good case for their performance scaling better with computation and data.
> > > > > We are looking forward to try out the proposed testbed. One possible advantage of using our  R2-D2 with the deeper architectures of their setup is that we can concatenate activations from multiple layers together without increasing the computational burden of the base-learner thanks to the Woodbury identity.

---

### Public Comment · (anonymous) · 2018-11-02
**What is the essential difference compared to training multiple models that share a pre-trained ConvNet (fine-tuning is allowed) providing input features?**

After reading this paper, I am confused about whether the proposed method is the same as a widely used technique, i.e., training multiple models (e.g., logistic regression) for different tasks based on shared input features provided by a pre-trained model (e.g., CNN), which can be fine-tuned. Although a minor difference here is that the tasks are sampled from a distribution of tasks rather than a fixed set (which follows a standard meta-learning setting), the used technique already exists and is well-known.

Since the proposed method is claimed to be a meta-learning approach that can quickly adapt to novel tasks, the training algorithm or the meta-learner should do something different for different tasks (i.e., adaptive to each specific task). However, The CNN remains the same for different tasks, and the closed-form solvers do not have any hyper-parameters changed with the task. I am not sure if it can be recognized as a meta-learning method. It might be more suitable to be categorized in multi-task learning, where models for different tasks share the same feature extractor (the CNN here).

Please correct me if I am wrong in the understanding of the essential idea of this paper. Thanks a lot!

---

> ### Public Comment · (anonymous) · 2018-11-06
> **shared parameters are optimized for Base test-set**
>
> IMO, shared parameters are optimized for Base test-set (Figure 1) instead of Base training-set, which is different than multi-task learning setup. ( I think AnonReviewer1 also raised similar issues...)
>
> And, I think authors missed a reference, which is very relevant.
> https://arxiv.org/abs/1806.04910

---

> > ### Author Response · Authors · 2018-11-08
> > **.**
> >
> > Thank you, this is a really nice paper. The bi-level optimization point of view is very insightful. Although their framework is very general, they seem to specialize it in the experiments using gradient descent for the inner loop, which is different from our closed-form solutions.

---

> ### Author Response · Authors · 2018-11-08
> **It's the procedure to generate the pre-trained convnet**
>
> > “I am confused about whether the proposed method is the same as … multiple models (e.g., logistic regression) for different tasks based on shared input features provided by a pre-trained model (e.g., CNN)”
>
> Thank you for participating in the discussion. This describes well only the behavior at test-time -- when facing a new task, a new regressor is learned based on pre-trained features (hence, different tasks will have different parameters). However, this leaves out a crucial detail: where does this pre-trained CNN come from?
>
> The standard approach is to use a CNN that was pre-trained on ImageNet or another task. However, there is no guarantee that the CNN features will transfer well to unknown tasks. In the case of few-shot learning, with only 1 or 5 training samples, fine-tuning will result in extreme over-fitting.
>
> Our training procedure (and indeed, all meta-learning methods for few-shot learning, such as MAML, SNAIL, etc) train the CNN features specifically to perform well on new, unseen tasks. “Performing well on unseen tasks” is formalized as achieving a low error after fine-tuning. This means that we have to back-propagate errors through the fine-tuning procedure, which can be SGD (MAML) or a ridge/logistic regression solver (ours). The end result is a CNN that is especially trained to be fine-tuned later under the same conditions; this differs substantially from standard pre-training.
>
> There is a nice, informal introduction to this (admittedly subtle!) distinction, that was written by the authors of MAML:
> https://bair.berkeley.edu/blog/2017/07/18/learning-to-learn/

---

> > ### Public Comment · (anonymous) · 2018-11-09
> > **Fine tuning on the test set of training tasks is not novel**
> >
> > Thanks a lot for your reply and explanation! I understand that the main novelty here is to apply fine tuning on the test set (of tasks sampled for training) in meta-learning, instead of on the training data of a single supervised learning task (as we normally did in supervised learning). However, I agree with AnonReviewer1: I do not think this work presents very original contributions. It applies the existing fine-tuning technique by following standard meta-learning setting, as many other meta-learning methods already did.
> >
> > Fine tuning is an existing technique that can be generally applied to different learning settings. The basic idea is to update a pre-trained model and continue to train it on new training instances. In supervised learning, each training instance is a data point, and the learning goal is to minimize the training error on each data point. In meta-learning, each training instance is an (n-way k-shot) classification task, and the learning goal is to minimize the validation/test error on the test set of each training task. Therefore, fine tuning in meta-learning should be applied to the test sets of training tasks (as this paper does). In fact, in meta-learning, any training happening on task-shared part (e.g., meta-learner or shared pre-trained model) should minimize the error/loss on the test sets of training tasks. However, these are all well-known facts, derived from the very early optimization formulation of "learning to learn" (although meta-learning becomes very popular topic very recently). So they are not the contributions of this paper.
> >
> > In addition, as the authors mentioned, many existing meta-learning methods use the same idea, the only difference here is that the base learner for each task changes to ridge/logistic regression model. But changing the model of base learners cannot be recognized as a novelty. Therefore, I think this is a successful application of existing technique, it re-explains how to do fine-tuning in meta-learning setting, but is not novel to me.

---

> > > ### Author Response · Authors · 2018-11-09
> > > **There is ample precedent in the few-shot learning literature for proposing new base learners as the main contribution.**
> > >
> > > Thank you.
> > >
> > > > “I understand that the main novelty here is to apply fine tuning on the test set (of tasks sampled for training) in meta-learning, instead of on the training data of a single supervised learning task (as we normally did in supervised learning).”
> > >
> > > Sorry but this is not claimed in the paper or in the answer above. Clearly, the overall training framework is not novel and it is common in the few-shot learning literature. In fact, we specifically wrote: “Our training procedure (and indeed, all meta-learning methods for few-shot learning, such as MAML, SNAIL, etc) ...”.
> > >
> > > The point of our previous comment was simply to clarify why different episodes correspond to different sets of parameters.
> > >
> > >
> > > > ““changing the model of base learners cannot be recognized as a novelty”
> > > We strongly disagree with the statement. This is exactly the nature of the contribution of most approaches for few-shot classification. For example, both MAML and prototypical networks use the same algorithm (SGD) in the external loop, while they vastly differ for the method used in the inner loop (SGD and nearest neighbour respectively).
> > >
> > > Our contribution is to use closed-form solvers such as ridge regression to tackle few-shot classification, which is novel in the literature and it is a non-trivial endeavor.
> > > As stated by AR2: “[it] strikes an interesting compromise between not performing any adaptation for each new task (as is the case in pure metric learning methods [e.g. prototypical networks]]) and performing an expensive iterative procedure, such as MAML or Meta-Learner LSTM where there is no guarantee that after taking the few steps prescribed by the respective algorithms the learner has converged.”
> > >
> > > Besides offering a trade-off with respect to existing techniques, our proposal also presents a significant practical value in terms of performance, as outlined in our experimental section.

---

> > > > ### Public Comment · (anonymous) · 2018-11-16
> > > > **Using a classical linear model as base learner is not novel to me**
> > > >
> > > > Thanks for your reply!
> > > >
> > > > > Clearly, the overall training framework is not novel and it is common in the few-shot learning literature.
> > > >
> > > > Thanks for the clarification! But if the "essential difference" (asked in my first post and answered in your previous reply) is not the contribution, it is hardly to tell the essential novelty of this method.
> > > >
> > > > > We strongly disagree with the statement. This is exactly the nature of the contribution of most approaches for few-shot classification.
> > > >
> > > > I do not agree with this statement. Simply replacing the base learner and following the standard meta learning/few-shot learning  scheme sounds not novel to me. The claimed adaptation capability comes from the standard meta-learning scheme, while the claimed efficiency comes from the closed-form solver. Both are well known and common for years.
> > > >
> > > > Yes MAML can be explained to be using SGD as base learner (but there are other more intuitive explanations), but they redesigned the learning procedure specifically for SGD, since SGD is a dynamic optimization algorithm rather than a model. Other meta-learning methods either proposes new algorithm or new model structure specifically for few-shot learning. BTW, I do not agree that "some papers propose their methods in the similar way, so our paper also presents contribution of similar novelty".
> > > >
> > > > >Our contribution is to use closed-form solvers such as ridge regression to tackle few-shot classification, which is novel in the literature and it is a non-trivial endeavor.
> > > >
> > > > Using closed-form solver for sure can converge faster than using deep neural networks or doing second order optimization (like MAML). But this is an advantage of the existing closed-form solvers. In addition, as mentioned in your reply and paper, the fine-tuning still needs to backpropagate the error from the closed form solver to the pre-trained deep CNN. Together they still compose a deep model whose last layer is the closed-form solver, and each epoch of the fine tuning might need heavy computation (**This has been also pointed out by Reviewer 1**). Then the advantage of using shallow model is not clear: you can always find a good trade-off between fine tuning a large/small backbone model and a complex/simple base learner. Besides, logistic regression does not have a closed-form solver so the title is somehow misleading.
> > > >
> > > > Overall, I agree that using closed-form solver of a shallow model might have some practical value, especially in the case when you use a very powerful pre-trained CNN as the backbone model. However, I am not convinced that this is a novel contribution.

---

> > > > > ### Comment · AnonReviewer2 · 2018-11-16
> > > > > **Re-using existing components in clever ways for new problems should be encouraged!**
> > > > >
> > > > > I respectfully disagree with the argument regarding lack of novelty. Indeed, the authors did not invent the meta-learning framework, and they did not invent ridge regression. Yet the two of them had not been combined before in this way, and this combination is evidently beneficial. It does seem like a natural idea, but if it was so obvious, how come it wasn't done before?
> > > > >
> > > > > It may be tempting to create complicated models to solve a problem, yielding "more novel" solutions. But this seems wrong if the same problem can be solved in a simpler way! I feel strongly that re-using existing components in clever ways that yield good results on new problems is an important contribution and should be encouraged.

---

> > > > > > ### Comment · AnonReviewer1 · 2018-12-08
> > > > > > **I respectfully disagree with Reviewer #2.**
> > > > > >
> > > > > > Merely combining ridge regression (trivial, and nothing novel) inside meta-learning is not sufficentlynovel in my opinion.
> > > > > >
> > > > > > In essence we agree to disagree. I request the AC to make a decision based on both our inputs.

---

### Author Response · Authors · 2018-11-26
**To all: Appendix now includes runtime analysis, 1-vs-all experiment and extended discussion**

We would like to thank both reviewers and anonymous commenters for their feedback and participation.
In light of the discussion, the Appendix of the paper has been updated:

* Section B offers a runtime analysis which reveals that R2-D2 is several times faster than MAML and almost as fast as a simple (fixed) metric learning method such as prototypical networks, while still allowing per-episode adaptation.
* Section A reports the accuracy of the 1-vs-all variant of LR-D2 (as suggested by AnonReviewer2), which is comparable with the one of R2-D2.
* Finally, Section C extends the discussion sparked here on OpenReview about a) the nature of our contribution b) the disambiguation with the multi-task learning paradigm .

---

### Public Comment · (anonymous) · 2019-01-08
**ICLR 2019 Reproducibility Challenge key findings**

We have carried out a  reproducibility analysis of this interesting paper on meta-learning. Some parameters and training methodologies, which would be required for full reproducibility, are not present in the manuscript at the time of writing:
- stride of the convolutional filters
- padding of the convolutional filters
- a clear stopping criterion (<-> "the error on the meta-validation set does not decrease meaningfully for 20k episodes"),

However, making reasonable assumptions, we were able to reproduce the most important part of the paper (R2D2) in TensorFlow and achieve similar results. We did not reproduce the LRD2 part of the paper, as we wanted to focus on the truly differentiable closed-form solver (R2D2). Most importantly, we were able to reproduce the increase in performance of the proposed method (with the given architecture) over some reproduced baseline results, which supports the conclusions in the original paper.

The different neural network architectures should be taken into consideration when comparing results. For example the MAML baseline of Finn et al. (2017) uses four convolutional blocks with [32, 32, 32, 32] filters, whereas this paper's four blocks employ a [96, 192,384, 512] scheme. Because of this we implemented R2D2 with both the architecture mentioned in the paper and the MAML baseline architecture. In our reproducibility report we show that when using the exact same baseline architecture as MAML, and standard training procedure, the improvement in performance of the proposed method is not clear.

Our full reproducibility report is available at: https://github.com/reproducibility-challenge/iclr_2019/blob/c53e6c1ea8d0e158f66b7d70681fa6ecde6a4f2b/papers/LCAX-HyxnZh0ct7/LCAX.pdf
Our codebase: https://github.com/ArnoutDevos/r2d2

---

> ### Author Response · Authors · 2019-02-11
> **Thank you for your contribution**
>
> We agree on the importance of making the research in ML (or any field) accessible and reproducible - we are glad that initiatives such as the reproducibility challenge exist.
>
> We are also glad that the authors were able to reproduce our main findings, despite not having had access to our implementation and using a different framework (we used PyTorch and we are working on finalizing the code release).
>
> In response to specific details of the report:
> - We agree that the details of stride and padding amounts are missing, and we will update the paper accordingly. This should also resolve the difference in feature dimension between our paper and the replication.
> - We believe that our sentence “Training is stopped when the error on the meta-validation set does not decrease meaningfully for 20,000 episodes” has been misinterpreted, as the authors say: “we opted to meta-train for a fixed 20k iterations”.
> What we meant is that we performed early-stopping if error does not decrease for a period of 20k episodes, not that we train for 20k episodes in total.
> Clearly, in this way the total number of training epoch varies, but we observed that generally it stops around 60k-80k episodes. We will make this point more clear in the camera ready. This also means that our results were obtained with longer training than in the replication.
> - Re the sentence “different neural architectures should be taken into consideration when comparing results” and direct comparison with MAML in general.
> This comment refers to the fact that we did not report results on a 32-channels embedding in our experiments, which is instead what MAML uses.
> However, we believe that our experiments already show that performance is not simply the result of a trivial increase in capacity.
> To demonstrate that, we reported both a) results of our method on a 64-channels embedding and b) results of three representative baselines (protonets, MAML and GNN) with our embeddings (the * in our tables).

---

### Author Response · Authors · 2019-02-22
**Camera-ready.**

The ICLR'19 camera-ready version of the paper has been uploaded.
Code available at https://github.com/bertinetto/r2d2

---

### Meta-Review · Area_Chair1 · 2018-12-14
**A closed form solver for the base learner is new in the meta-learning literature, and the experiments are sufficiently carried out to show its effectiveness.**

**Confidence:** 4
**Recommendation:** Accept (Poster)

**Metareview:**

The reviewers disagree strongly on this paper. Reviewer 2 was the most positive, believing it to be an interesting contribution with strong results. Reviewer 3 however, was underwhelmed by the results. Reviewer 1 does not believe that the contribution is sufficiently novel, seeing it as too close to existing multi-task learning approaches.

After considering all of the discussion so far, I have to agree with reviewer 2 on their assessment. Much of the meta learning literature involves changing the base learner *for a fixed architecture* and seeing how it affects performance. There is a temptation to chase performance by changing the architecture, adding new regularizers, etc., and while this is important for practical reasons, it does not help to shed light on the underlying fundamentals. This is best done by considering carefully controlled and well understood experimental settings. Even still, the performance is quite good relative to popular base learners.

Regarding novelty, I agree it is a simple change to the base learner, using a technique that has been tried before in other settings (linear regression as opposed to classification), however its use in a meta learning setup is novel in my opinion, and the new experimental comparison regression on top of pre-trained CNN features helps to demonstrate the utility of its use in the meta-learning settings.

While the novelty can certainly be debated, I want to highlight two reasons why I am opting to accept this paper: 1) simple and effective ideas are often some of the most impactful. 2) sometimes taking ideas from one area (e.g., multi-task learning) and demonstrating that they can be effective in other settings (e.g., meta-learning) can itself be a valuable contribution. I believe that the meta-learning community would benefit from reading this paper.